# Decomposition Analysis of Virtual Water Outflows for Major Egyptian Exporting Crops to the European Union

**Samaa Mohy [1,2,*]**, **Khadija El Aasar [1]** and **Yasmin Sakr [1]**

[1] Department of Economics, Faculty of Economics and Political Science, Cairo University, Giza 12613, Egypt
[2] Department of Economics, Faculty of Commerce-English Section, Benha University, Benha 13518, Egypt
* Correspondence: samaa_zaki2014@feps.edu.eg

**Abstract:** International trade can spur economic growth, but it can also deplete the water resources needed to produce traded goods. This is crucial for Egypt as a lower-middle income country where boosting agricultural exports is encouraged to promote the sustainable agriculture development strategy. The objective of this paper was to quantify Egypt's virtual water flows contributing to agricultural trade with one of its main trading partners, the European Union. We considered calculating virtual water of exports since 2001 as it represents Egypt-EU's implementation of the association agreement. We focused on the five governorates of the Nile Delta. These governorates are major producers of the five major crops exported to the European Union. This study used long-term trade trends, and changes in crop composition to analyze the implications for virtual water outflows and economic water use efficiency. By decomposing the virtual water of exports, we were able to identify the trend of virtual water outflows and the factors affecting this trend. From both an economic and water perspective, our results suggest that adopting a policy aimed at saving water resources at the national level and focusing on high-yield exports at the international level will promote the development agenda of Egypt.

**Keywords:** evapotranspiration; water footprint; CROPWAT 8.0; virtual water trade; economic water efficiency; agricultural trade; decomposition analysis




## 1. Introduction

Water and agriculture are key factors in economic growth [1]. Despite being a renewable resource, water is a limited natural resource [2]. These limitations are demonstrated in Egypt by its fixed share of the Nile River (55.5 million $m^3$/year), scarce rain, minimal contribution from underground water and desalination, and the increased recycling of agricultural drainage water [3].

The Nile has provided water and irrigation for 11 African countries: Burundi, the Democratic Republic of Congo, Egypt, Eritrea, Ethiopia, Kenya, Rwanda, South Sudan, Sudan, Tanzania, and Uganda (see Figure 1). It is crucial for the survival of more than 260 million people living in riparian countries. Egypt and Sudan are entirely dependent on Nile water. With Ethiopia announcing, in 2011, the construction of the Grand Ethiopian Renaissance Dam (GRED) across the Blue Nile River and global warming concerns such as changes in precipitation patterns and depletion of groundwater, the regional water demand is expected to rise [4,5]. A reduction in water supplies caused by GERD will affect Egypt's water resources, since most of them are recharged through the Nile [6]. This will in turn impact the agriculture sector which consumes 85% of Egypt's water supplies and is the main driver of socio-economic growth in Egypt [7]. Thus, estimating actual water consumption for crops, is essential for water resource management [8].

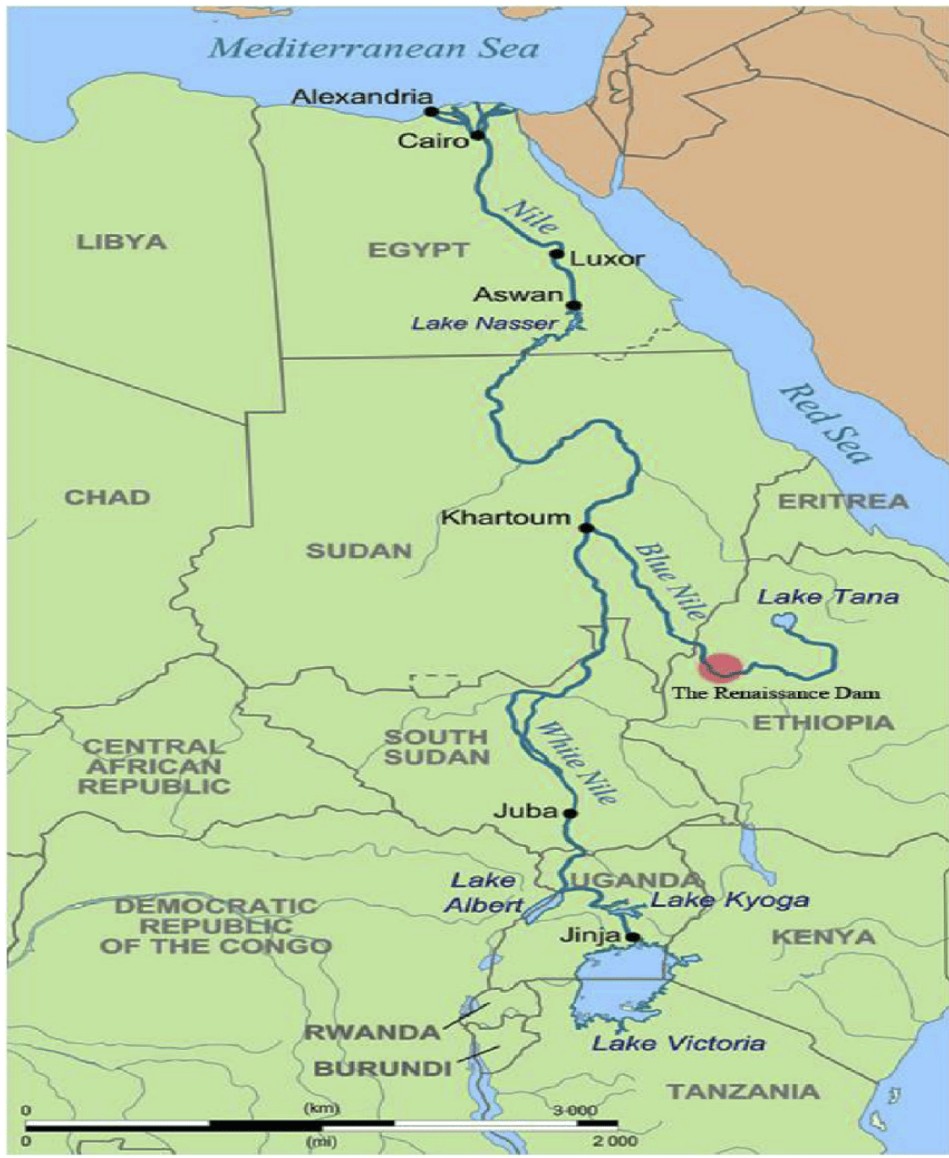

**Figure 1.** Map of the Nile basin countries.

Furthermore, uneven distribution of water resources, caused by the natural hydrological cycle and different climatic zones, has resulted in a socio-economic companion of human-induced cycle through water virtually embedded in products. Global trade networks allow virtual water to flow from surplus areas to shortage areas, sometimes in reverse [9]. In evaluating water resources, Tony Allen (1997) referred to the concept of "virtual water" (VW) as "embedded water" to describe the water volume required to grow, produce, and package agricultural commodities. Hoekstra and Hung (2002) developed the water footprint (WF) concept as a way to quantify these VW flows in international food trade [10–12]. Based on VW measures, it was necessary to find a way to analyze the correlation between human consumption in one place and freshwater appropriation in another. This led to the development of the concept of WF [13]. WF has a broader perspective than VW. It offers consumption-based indication of water use that tracks water consumption associated with different consumption items throughout the entire supply chain whether internal (consumed domestically) or external (imported in a virtual form) [14]. By using the WF, we can relate or link or connect VW flows to crop water use, reflecting the crop pressure on water resources [15]. Through international trade between countries, inflows of virtual water reduce the demand for domestic water resources while outflows increase it [16]. Index decomposition analysis (IDA) divides the change in water consumption

into predefined factors like water footprint, virtual water trade, economics, and water efficiency improvements. These results enable policymakers to understand major trends driving water demand, quantify water savings from water efficiency use, and forecast future water demand. This allows integrated assessments of water, environment, and economic concerns [17]. A methodology for decomposing VW flows, index decomposition analysis, has recently been applied by Schwarz et al. and Duarte et al. [18,19].

A global study of international VW flows for Hoekstra and Makonnen indicated that crops and crop products account for the majority (76%) of VW flows between countries, with animal and industrial products contributing 12% each [20]. On a global scale, many studies have examined the relationship between trade and water resources, emphasizing on agricultural trade [21–23]. According to Wahba et al., Egypt has witnessed a change in the production and trade of water-intensive crops over the past 50 years. As an example, Egypt's wheat production increased by 570% between 1961 and 2015, accompanied by 1456% growth in net imports. Similarly, Egypt's rice production increased by 322% during the same period, while its net rice exports increased by 590% [24]. Using hydrological and economic perspectives, Khalil et al. quantified the virtual water trade flows for several crops in Egypt between 2008 and 2012. They compared their virtual water trade balances with national water requirements and availability. Their recommendation was to stop exporting rice and sugarcane, cover national consumption and plant rice in Lower Egypt and sugarcane in Upper Egypt. This is because they have a smaller footprint and are more cost effective. They recommended stopping the import of wheat and maize, increasing their yield to cover any deficit, and expanding the planted area in Lower Egypt [25].

Although many studies have estimated the water footprint for agricultural crops and virtual water trade for Egypt [26–28], none of them focused on identifying virtual trade between Egypt and the EU and the main driving factors behind changes in VW outflows. With over 25% of Egypt's trade, the EU is the nation's most significant trading partner. Since the free trade agreement (FTA) went into effect, bilateral commerce in products has risen almost threefold, rising from €8.6 billion in 2003 (the year before the Association Agreement went into effect) to €24.5 billion in 2020. Fuels, mining products, chemicals, and agricultural goods are Egypt's main exports to the EU [29].

Rapid economic growth in countries with limited water resources, like Egypt, requires planning beyond the local water supply. The paper seeks to introduce a recently used analysis for virtual water trade, an index decomposition analysis (IDA), in order to identify trends and sources of change in virtual water outflows with the EU, one of Egypt's major trading partners for a long period of time. This analysis will be preceded by a bottom-up approach to calculate water footprint through average monthly evapotranspiration data for the whole period. Then, we will link these crop water intensities with the volume of bilateral trade provided by CAMPAS between Egypt and the EU. As the key compositional factors are established, IDA will identify the main driving factors for changes in virtual water outflows. The findings will help policy makers to identify virtual water driving factors and adjust trade structures to limit them.

## 2. Materials and Methods

### 2.1. Estimation of Reference Evapotranspiration (ET₀) and Crop Coefficient (Kc)

For computing crop water requirements, the FAO introduced the concept of "reference crop evapotranspiration" to study the evaporative demand of the atmosphere at a specific location and time of the year independently of crop type, crop development and management practices. An estimate of $ET_0$ is the first step in estimating actual evapotranspiration $ET_a$. The only factors affecting $ET_0$ are climatic parameters and can be calculated from weather data based on the FAO Penman-Monteith equation [30]:

$$ET_0 = \frac{0.408\,\Delta\,(R_n - G) + \gamma\left(\frac{900}{T+273}\right)U_2\,(e_s - e_a)}{\Delta + \gamma(1 + 0.34U_2)} \tag{1}$$

where $ET_0$ reference evapotranspiration [mm day$^{-1}$], $Rn$ net radiation at the crop surface [MJ m$^{-2}$ day$^{-1}$], $G$ soil heat flux density [MJ m$^{-2}$ day$^{-1}$], $T$ mean daily air temperature at 2 m height [°C], $U_2$ wind speed at 2 m height [m s$^{-1}$], $es$ saturation vapor pressure [kPa], $e_a$ actual vapor pressure [kPa], $e_s - e_a$ saturation vapor pressure deficit [kPa], $\Delta$ slope vapor pressure curve [kPa °C$^{-1}$], $\gamma$ psychrometric constant [kPa °C$^{-1}$].

Integrating specific crop characteristics (e.g., planting and harvesting dates, crop resistance, albedo, crop height, and critical depletion) and soil characteristics of the specific location with the climatic data of the reference evapotranspiration under certain management and environmental conditions, we can get the adjusted crop evapotranspiration for a specific crop at a specific time and location by the following equation [31]:

$$ET_{c,\,adj} = K_c \times K_s \times ET_0 \tag{2}$$

where $ET_{c,\,adj}$ the adjusted crop evapotranspiration under non-standard conditions [mm day$^{-1}$], $K_c$ adjusted crop coefficient, $K_s$ water stress coefficient, and $ET_0$ reference evapotranspiration [mm day$^{-1}$].

*2.2. Virtual Water Trade*

International virtual water trade accounts for one third of world water withdrawals, of which 43% is food trade. This highlights the importance of assessing virtual water trade of both food products and crops [32]. In this study, virtual water of exports of five agricultural crops to the EU are estimated based on water footprint estimation, which is in turn calculated from crop water use ($CWU_c$) and yields ($Y_c$). These calculations follow the method described by Hoekstra et al. [33].

The evapotranspiration (mm) over the complete growing season is used to calculate $CWU_c$ and is converted to m$^3$/hectare applying the factor 10. $CWU_c$ then divided by the yield to obtain the water footprint for each crop ($c$).

$$CWU_c = 10 \times \sum_{t=1}^{lgp} ET_c \tag{3}$$

$$WF_c = \frac{CWU_c}{Y_c} \tag{4}$$

To calculate the annual virtual water outflows between Egypt and the EU, we multiplied international crop trade outflows by the associated virtual water content (i.e., water footprint). The latter is affected by the specific water demand for the crop in the exporting country. The virtual water trade can therefore be calculated as follows [34]:

$$\text{VWX (t)} = \sum_p X_q(p, t) \times WF(p, t) \tag{5}$$

Using annual trade values and associated VW volumes, we calculate economic water efficiency of exports (EWEX). It gives us an idea of how much money is earned on each unit of VW outflow per year. Annual export values are divided by respective VW volumes to calculate this ratio as follows [35]:

$$\text{EWEX (t)} = \frac{X_v(t)}{VWX(t)} \tag{6}$$

*2.3. Refined Laspeyres Method*

Sun 1998 suggested a complete decomposition analysis where the residual term is shared among the predefined factors evenly. That means that residuals generated by two factors are allocated equally, while residuals generated by three factors are equally distributed. This is known as the refined Laspeyres method [36,37].

The Laspeyres index uses weights derived from values in some base year to calculate the percentage change in some aspect of a group of items over time. This method isolates the impact of a factor by allowing variables related to this factor to change from one year to the next while keeping variables related to the other factor constant. There is, however, a problem with this method that the number of terms required to allocate the residual increases exponentially if more than three factors are considered, but our analysis is unaffected by this limitation [38].

Decomposition of virtual water outflows for each crop

$$VWX_c = Q_x \times \frac{Q_c}{Q_x} \times \frac{VWX_c}{Q_c} \tag{7}$$

where $VWX_c$ the virtual water exports for each crop, $Q_x$ the total annual quantity exported of all crops (in tons), $Q_c$ the annual quantity exported for each crop. Decomposition of virtual water outflows for all five crops (which are considered among the most important agricultural exports to the EU).

$$VWX = \sum_c Q_x \times \frac{Q_c}{Q_x} \times \frac{VWX_c}{Q_c} \tag{8}$$

where $VWX$ refers to the total virtual water exports or outflows for all five crops, $\frac{Q_c}{Q_x}$ measures the share of crop $c$ in exports in (%), and $\frac{VWX_c}{Q_c}$ indicates the water intensity of production per unit of $c$, i.e., the water footprint (m$^3$/ton). Denoting $\frac{Q_c}{Q_x}$ as $S_c$ and $\frac{VWX_c}{Q_c}$ as $WF_c$. Each factor's annual change can be expressed as follows:

$$\Delta Q_x = Q_x^t - Q_x^{t-1} \tag{9}$$

$$\Delta S_c = S_c^t - S_c^{t-1} \tag{10}$$

$$\Delta WF_c = WF_c^t - WF_c^{t-1} \tag{11}$$

The variation in virtual water exports can be decomposed into three effects. First, the scale effect (SE), also known as the activity or total effect, shows how much the economy's expansion in exports has increased virtual water outflows. VWX rises linearly as $Q_x$ increases if the scale effect is the dominant influence. Second, the composition effect (CE), or variations in the content of exported crops over time, illustrates the impact of structural changes in trade on VW outflows. VWX will be less if the economy focuses on less water-intensive crops. Third, the water intensity effect (WE), also called direct effect, connects variations in VW outflows to changes in the crop water footprint over time. Therefore, working on the global warming issue and changing the irrigation techniques can reduce water intensity. The last effect shows the importance of technical advancements [39,40].

The scale effect:

$$SE = \sum_c \Delta Q_x \times S_c^{t-1} \times WF_c^{t-1} + \frac{1}{2}\Delta Q_x \left[\Delta S_c \times WF_c^{t-1} + S_c^{t-1} \times \Delta WF_c\right] + \frac{1}{3}\Delta S_c \times \Delta WF_c \times \Delta Q_x \tag{12}$$

The composition effect:

$$CE = \sum_c \Delta S_c \times WF_c^{t-1} \times Q_x^{t-1} + \frac{1}{2}\Delta S_c \left[\Delta WF_c \times Q_x^{t-1} + WF_c^{t-1} \times \Delta Q_x\right] + \frac{1}{3}\Delta S_c \times \Delta WF_c \times \Delta Q_x \tag{13}$$

The water-intensity effect:

$$WE = \sum_c \Delta WF_c \times S_c^{t-1} \times Q_x^{t-1} + \frac{1}{2}\Delta WF_c \left[\Delta S_c \times Q_x^{t-1} + S_c^{t-1} \times \Delta Q_x\right] + \frac{1}{3}\Delta S_c \times \Delta WF_c \times \Delta Q_x \tag{14}$$

We use the trade data collected and the VW outflows dataset calculated as described above. The annual effects are calculated for various time periods, and we compare the relative contributions of SE, CE, and WE to changes in VW exports for each crop.

$$\alpha_c = \frac{SE_{effect}}{VWX_c^t - VWX_c^{t-1}} \tag{15}$$

$$\beta_c = \frac{CE_{effect}}{VWX_c^t - VWX_c^{t-1}} \tag{16}$$

$$\delta_c = \frac{WE_{effect}}{VWX_c^t - VWX_c^{t-1}} \tag{17}$$

where $\alpha_c$, $\beta_c$, and $\delta_c$ denotes the relative share of SE, CE, and WE effect to the virtual water outflows, respectively. By adding up each effect across all crops and expressing this number as a percentage of the annual change in total agricultural VW exports, the effects for the crops representing the agricultural sector are estimated [41].

*2.4. Selected Crops*

Onions, potatoes, and tomatoes are important cash crops in Egypt. They generate a significant amount of income. As the largest vegetable crop in both area and production, tomatoes account for approximately 150 thousand hectares or 18.3% of the total vegetable area, producing 6.3 million tons in 2021. Egypt ranks fifth globally in tomato production after China, India, Turkey, and the U.S. Tomatoes are available all year round in Egypt due to its ability to produce them in all governorates and all seasons. In lower Egypt, summer tomatoes are often grown in April and harvested in July. Onions are among the most cultivated vegetables after tomatoes, with an estimated harvested area of 5.2 million hectares, and a production of 100 million tons globally. Among the ten largest onion producers in the world, Egypt occupies the fifth position in terms of area and ninth in terms of productivity. In the Egyptian trade, onions are an important crop, with more than 85 thousand hectares of cultivated land and a self-sufficiency rate of 120.25%. In 2021, Egypt produced about 3.5 million tons of onions. It is a winter crop that is planted in Al-Gharbiya in October and harvested in April. Potato come from the Solanaceae family. As of 2021, Egypt produced 6.2 million tons of potatoes from a cultivated area of 211 thousand hectares, with a self-sufficiency rate of 111.4%. Winter potatoes are cultivated in October in Delta region, and harvested in February, with most of it being exported early to Europe [42,43].

Agriculture was a key component of Egypt's development policies during the 1960s, with cotton and rice, at the top of Egyptian exports, constituting about 80 percent of all exports. During this period, the policy makers aimed to level the playing field by controlling crop schedules, crop area allocations, quotas, and subsidizing consumer prices. Farmers became frustrated with this interventionist institutional structure, and subsequently resulted in yields being dropped, cropping patterns distorted, crop self-sufficiency gaps widened, food subsidies burdened the government, urban-rural income gaps widened, and a reduction of exports affected the cotton and rice export market. Due to the agricultural policy of the 1960s and 1970s favoring wheat and maize over cotton in order to meet local food demands, cotton exports and cultivation declined. From 1986 through the 1990s, interventionist policies were reversed by liberalizing economic reform policies, giving the private sector a larger role in agriculture, and reducing government involvement. Despite agricultural policy reforms from 1987 to 2002, there was still a decline in cotton yields and exports, followed by a broader reform that included privatization of public firms. Rice and cotton are summer crops that complement wheat cultivation in crop rotations, which explains the increase in rice production as cotton production declines. Rice is considered one of the main cereal crops in Egypt, which occupies a large percentage of irrigated land [44,45].

Rice, cotton, potatoes, tomatoes, and onions are five of the major export crops exported to the EU during this study period. We assume that most of the exported quantities came from the winter harvest of potatoes and onions, as well as the summer harvest of tomatoes. This assumption was based on the significantly greater production of crops during these seasons.

### 2.5. Study Area

Egypt is in the northeastern corner of Africa and has an area of approximately one million km². The country is bordered by the Mediterranean Sea to the north, the Gaza Strip and the Red Sea to the east, Sudan to the south, and Libya to the west [46]. The agricultural areas of Egypt have been divided into five geographical regions based on their characteristics. The five regions include Upper Egypt, Middle Egypt, Middle Delta, Eastern Delta, and Western Delta. As shown in Figure 2, Upper Egypt is formed by the following area/districts; Asyut, Suhag, Qena, Aswan, and the New Valley governorates. Giza, Beni Suef, Fayyum, and Minya form Middle Egypt, while Al-Qalyoubiya, Al-Menoufiya, Al-Gharbiya, Al-Daqahlia, Kafr El-Sheikh, and Dumyat governorates make up the Middle Delta. Eastern Delta governorates includes Port Said, Ismailia, Suez, Northern Sinai and Southern Sinai, and the Western Delta governorates includes Al-Behera, Alexandria, Al-Nubariya, and Matrouh governorates [47]. Egypt's agricultural production is centered on the Nile Delta in the north, overlooking the Mediterranean Sea with a 240 km coastline stretching from Alexandria in the west to Port Said in the east. There are about 3.8 million hectares of agricultural land in Egypt (old lands and new lands) covering about 4% of Egypt's total area. Agriculture land in the Nile Delta occupies about 2% of Egypt's land area [48].

In this study, the selected crops were grown in Kafr El-Shaykh, Al-Daqhliya, Al-Nubariya, Al-Gharbiya, and Al-Beheira governorates. According to CAMPAS, in 2021, about 99.5% of the production of rice was cultivated in the Nile Delta governorates, which represented 99.5% of Egypt's total rice production. Al-Daqhliya governorate was ranked first place for the production of 1.2 million tons (27.94%) that year. About 93% of cotton is produced in Egypt's Nile Delta governorates, with Kafr El-Shaykh governorate having the highest production of 101 thousand tons (33.3%). These governorates are also home to various vegetable productions. Al-Nubariya governorate ranks first in tomato production with 1.2 million tons (19.2%). Al-Gharbiya produces 920 thousand tons (25.7%) of onions. Al-Beheira governorate dominated potato production until 2019 by producing 758 thousand tons (14.6%). The cultivation of these field crops in each governorate requires a considerable percentage of water requirements as seen in Table 1 [49].

**Table 1.** Highest field crop area and production.

| Field Crops | % of Field Crops' Area of Total Field Crops Area in Egypt | % of Field Crops' Production of Total Field Crops Production in Egypt | % of Field Crops' Water Requirements of Total Field Crops Water Requirements in Egypt |
|---|---|---|---|
| Cotton (Kafr El-Shaykh) | 35.8 | 29.5 | 39.1 |
| Rice (Al-Daqhliya) | 26.7 | 27.9 | 24.8 |
| Tomatoes (Al-Nubariya) | 25.7 | 19.2 | 24.2 * |
| Onions (Al-Gharbiya) | 20.7 | 25.7 | 25.9 |
| Potatoes (Al-Beheira) | 15.3 | 14.6 | 16.5 * |

* Indicative figure (the data available is for all vegetable crops).

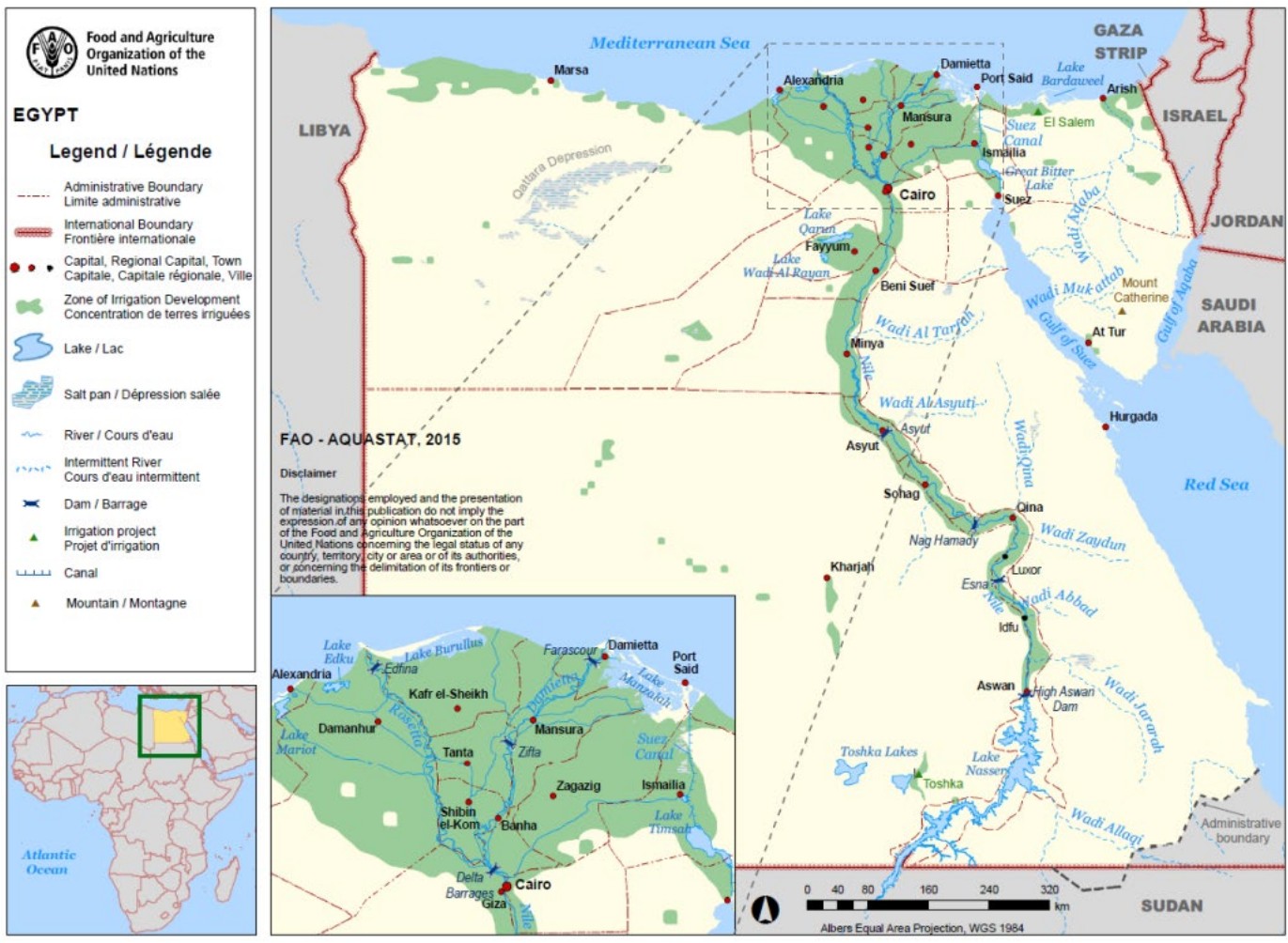

**Figure 2.** Map of Egypt.

*2.6. Collected and Measured Data*

From 2001 to 2021, the average monthly measured reference evapotranspiration data $ET_0$ (mm/day) and some of the crop data were collected from the central laboratory for agricultural climate and the ministry of agriculture and land reclamation MALR. Yield and trade data were collected from CAMPAS, FAO statistics, and the international trade center. Soil parameters and some of the crop parameters were taken from the default values in the CROPWAT 8.0 software provided by FAO. Crop evapotranspiration $ET_c$ (mm/dec), crop water use CWU (m$^3$/hectare), water footprint WF (m$^3$/ton), and virtual water of exports VWX (m$^3$/year) were calculated yearly for the whole period for the selected crops. Decomposition analysis was performed with the trade data collected and VW outflows dataset calculated.

**3. Results**

*3.1. Exports*

Exports of some agricultural crops have increased, while others fluctuated and decreased since 2001. Exports of potatoes, tomatoes, and onions have fluctuated and increased, while exports of rice and cotton have decreased significantly as indicated in Figure 3.

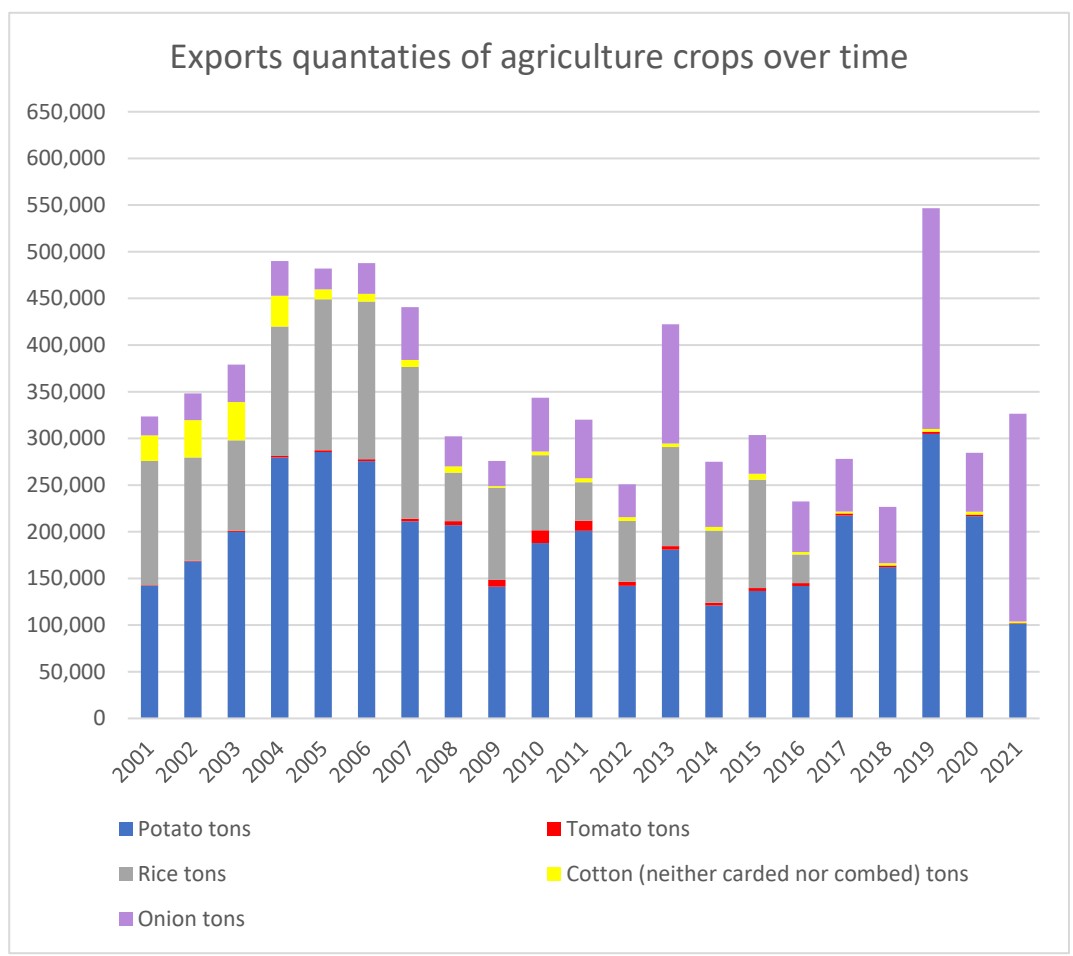

**Figure 3.** Exports quantities of agriculture crops to the EU over time.

In Egypt, potatoes, tomatoes, and onions are the main exported vegetables. potatoes constitute the first and largest horticultural export in Egypt, with 216 thousand tons exported to Europe in 2020, which represents 33.5% of Egypt's total potato exports (796 thousand tons). The overall increase in potato exports was due to increasing the yield, which was accompanied by expansion in the cultivated crop area.

Most of potato production (93%) is consumed domestically. Egypt has been found to possess a comparative advantage in potato exports to the EU markets. The exports of potatoes to the EU are, however, declining in some periods due to the lack of an efficient export plan, as well as the low export price competitiveness and disease outbreaks [50].

Despite being the largest vegetable crop in both area and production, with 6.8 million tons on average between 2016 to 2020, the average amount of tomatoes exported from Egypt was only 600 tons in 2001. In 2019, exports of Egyptian fresh tomatoes reached 2300 tons.

Onion is one of Egypt's most significant agricultural export crops, valued at $155.36 million in 2020. During the period (2001–2021), Egyptian onion exports to these countries have tripled from 20 thousand tons to nearly 63 thousand tons on average, with a boom in 2013, 2019 and 2021 in the total Egyptian onion export, reflecting on EU exports as shown in Figure 3. This highlights the positive effect of the EU trade agreements with Egypt on onion exports and the competitiveness of the Egyptian onion on trade markets.

Despite Egypt's comparative advantage in rice, its strategic and significant importance, and its high percentage of cultivated land, these advantages have been lost. As shown in Figure 4, rice exports have grown from 47 thousand tons in 2000 to 162 thousand tons in 2007. Despite the sharp decline in 2007 due to the ban being released in 2012, the quantity

of rice exported has not reached the boom recorded before 2008. This can be attributed to the conservative water plan Egypt started to adopt back in 2009.

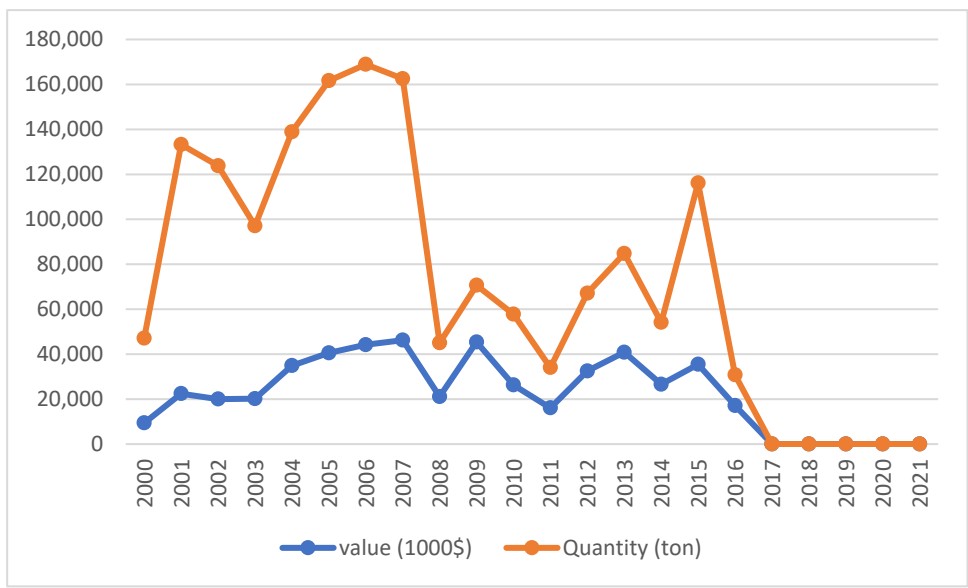

**Figure 4.** Value and quantity of Egyptian rice exports to the EU during (2000–2021).

The quantity of Egyptian cotton exported to the EU has fallen from 40 thousand tons in 2001 to 1.6 thousand tons in 2021. This can be attributed primarily to the sharp reduction in cotton cultivated land from 307 thousand hectare in 2001 to nearly100 thousand hectare in 2021. The arbitrary low yield of cotton was one of the major reasons for farmers to shift to higher yield crops, which involves higher productivity and profit, especially after liberalization policies at the beginning of the 1990s.

We split the period of 2001–2021 into three phases. A reduction in the exports of the selected crops was primarily caused by the drop in potato, and rice trade as depicted in Table 2, which represented 86%, 78%, and 65% of total exports in the first, second, and third phase, respectively. The cotton crop showed a very poor trend over time due to the reduction in planted area in favor of rice. During all three phases, the share of cotton is insignificant, and it continued to decline. Although onion and tomato exports have increased over time, they represent a small portion of Egyptian exports to the EU compared to other selected crops. Potato and rice were the most exportable crops in the first two phases due to their price competitiveness, large cultivatable areas, and high production, however the export of rice dropped sharply beyond onion exports in the third phase because of the trade ban imposed by the government. Onion showed a significant increase in quantity exported to the EU throughout the three phases.

**Table 2.** Average quantity and percentage of Egyptian Exports and associated Virtual Water to the EU.

| Titles | Average Egyptian Exports to the EU | | | | | | Average Virtual Water of Exports to the EU | | | | | |
|---|---|---|---|---|---|---|---|---|---|---|---|---|
| | (2001–2007) | | (2007–2014) | | (2014–2021) | | (2001–2007) | | (2007–2014) | | (2014–2021) | |
| | Tons | % | Tons | % | Tons | % | (10⁵ m³) | % | (10⁵ m³) | % | (10⁵ m³) | % |
| Potato | 223,134 | 53 | 168,716 | 54 | 182,995 | 58 | 334 | 10.4 | 223.9 | 14.8 | 256.8 | 31.6 |
| Cotton | 23,990 | 6 | 4046 | 1 | 3022 | 1 | 942.3 | 29.4 | 216.5 | 14.3 | 153.1 | 18.8 |
| Rice | 138,979 | 33 | 74,461 | 24 | 20,981 | 7 | 1877.5 | 58.5 | 977.4 | 64.7 | 276.4 | 34 |
| Tomato | 1526 | 0 | 6019 | 2 | 1991 | 1 | 3.1 | 0.1 | 11.3 | 0.7 | 3.6 | 0.4 |
| Onion | 33,983 | 8 | 58,886 | 19 | 104,931 | 33 | 51.6 | 1.6 | 80.9 | 5.4 | 123.6 | 15.2 |

### 3.2. Virtual Water of Exports and Water Footprint Assessment

When we compared the export trade quantities (Figure 3), we noticed that VWX for the selected agricultural crops dropped from 326 million m³ to 45 million m³ between 2001 and 2020. This highlights the effects of the exports fall on the decline in virtual water outflows for water-intensive crops as depicted in Figure 5. VW outflows are concentrated in rice and cotton crops, contrary to the product composition of trade quantities. During the period of 2001–2007, exports of rice and cotton constituted 39% of agricultural exports to the EU, which dropped to 8% during 2014–2021. Their virtual water outflows during the same periods represents 88% and 53% for the first and last period, respectively (Table 2). This illustrates the drop in the amount of virtual water outflows during the second and third period due to declining exports of rice and cotton.

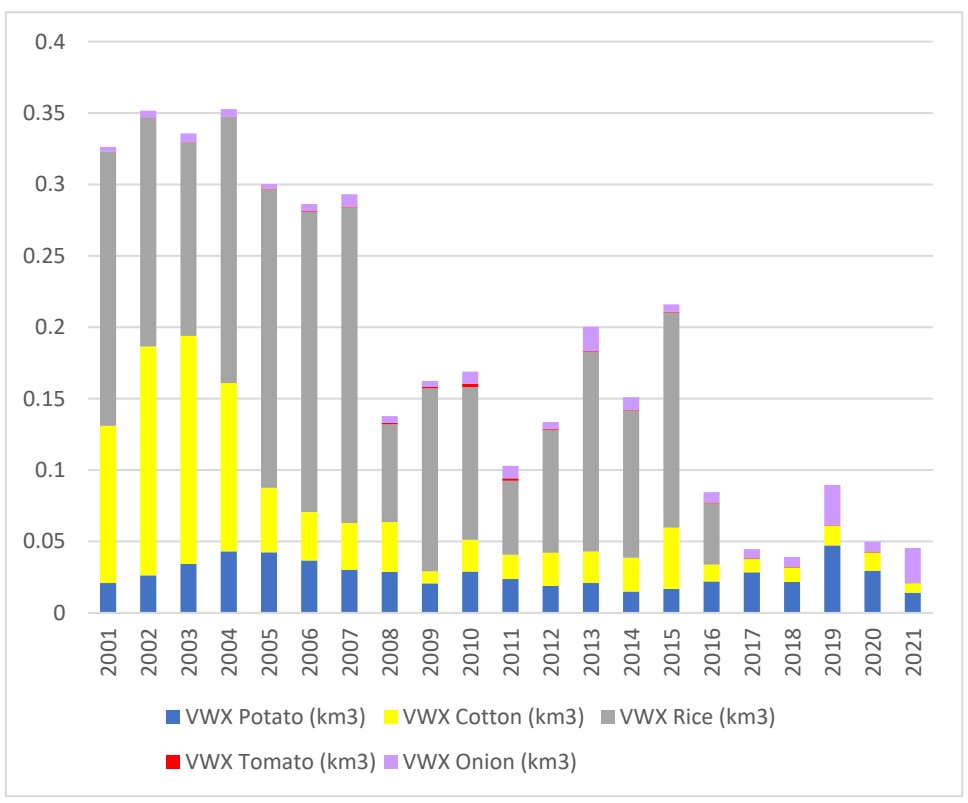

**Figure 5.** Virtual water exports of Egyptian crops to the EU from 2001 to 2021.

On the other hand, according to Equation (4), periods of high crop yields reflect a decrease in the water footprint. With the improvement of cultivation methods and the adoption of modern agricultural technologies, crop yields increase, and water footprint decreases. Accordingly, the average water footprint for onion crop decreases as its average yield increases during the three periods as shown in Table 3.

**Table 3.** Average crop Yield and water footprint for three periods.

| Titles | Average Crop Yield | | | Average Water Footprint for Crops | | |
|---|---|---|---|---|---|---|
| | (2001–2007) | (2007–2014) | (2014–2021) | (2001–2007) | (2007–2014) | (2014–2021) |
| | Ton/hectare | Ton/hectare | Ton/hectare | (m³/ton) | (m³/ton) | (m³/ton) |
| Potato | 22.14 | 26.39 | 23.81 | 150.48 | 132.68 | 138.94 |
| Cotton | 2.72 | 2.11 | 2.47 | 4020.62 | 5348.52 | 4681.06 |
| Rice | 10.09 | 10.52 | 9.71 | 1360.45 | 1309.82 | 1446.05 |
| Tomato | 29.45 | 36.23 | 36.30 | 207.57 | 172.39 | 171 |
| Onion | 32 | 37 | 40 | 152.57 | 139.40 | 120.68 |

A significantly high WF was observed for cotton followed by rice due to the massive CWU of both crops and the sharp decline in cotton yield. Potatoes, onions, and tomatoes, on the other hand, had a substantially lower water footprint, as shown in Figure 6.

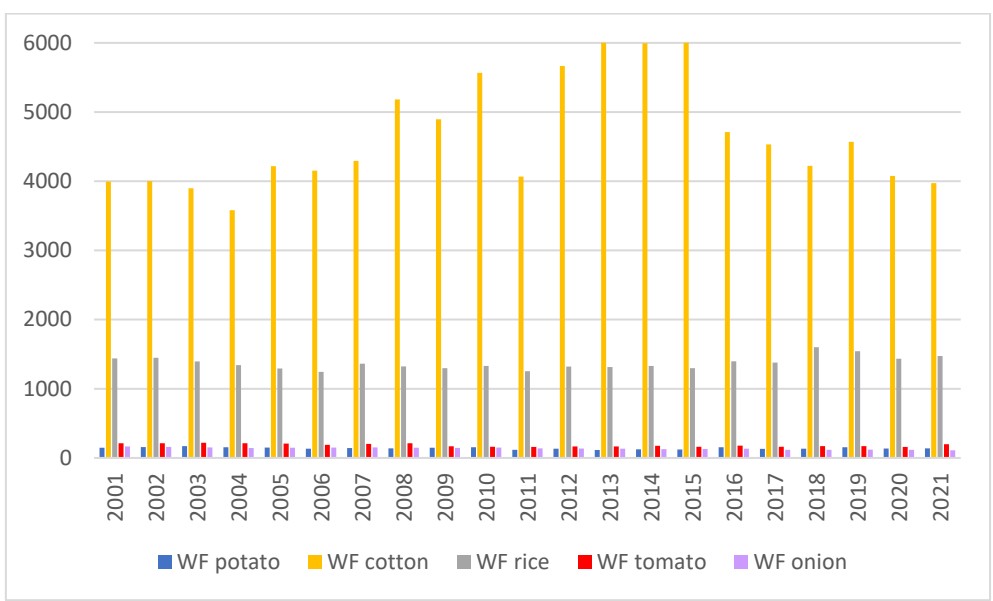

**Figure 6.** Water footprint of the five crops over the period (2001–2021).

A closer look at the crop yield over the study period is required to determine the reasons behind the increase in WF. Figure 7 depicts crop yield values (ton/hectare) for the studied crops, revealing a very low yield for cotton, a slightly higher yield for rice and substantially higher yields for tomatoes, onions, and potatoes.

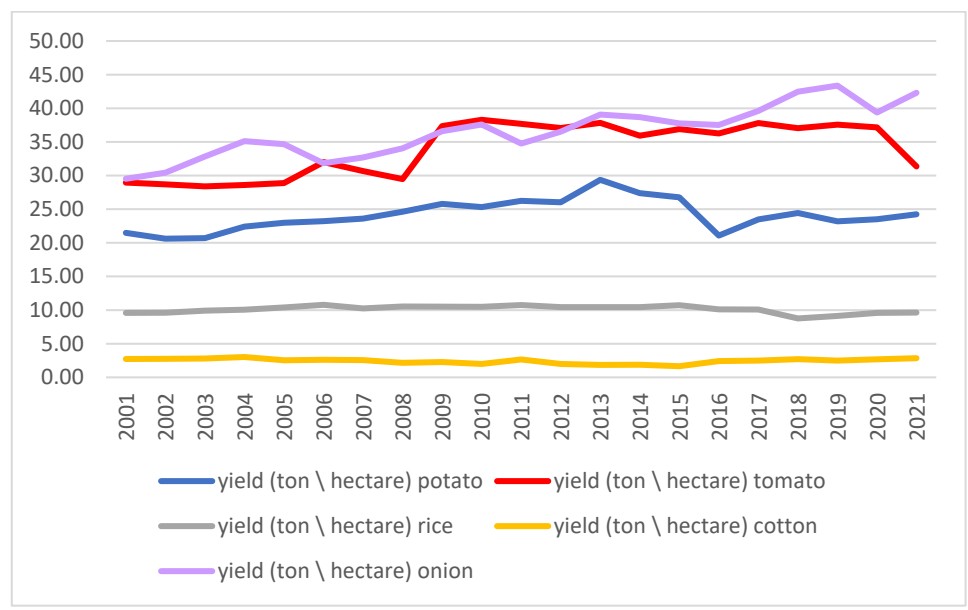

**Figure 7.** Crop yield over the period (2001–2021).

### 3.3. Economic Water Effeciency of Trade

The economic water efficiency of exports (EWEX) has improved over the previous decade for tomatoes, onions, and potatoes, while it remained the same and decreased for cotton and rice respectively. This can be explained by the reduction in rice and cotton export values. A lower growth rate of export values for potatoes and onions reflects the

medium growth of their economic water efficiency, while increasing the value of tomatoes contributes largely to the higher growth rate of the economic water efficiency and a different product composition of trade during that time. A peak value of 15.76 USD/m³ for exports of tomatoes in 2018 was demonstrated. In Table 4, EWEXs are shown for different crops over the whole study period. Exports of tomatoes had a higher economic water efficiency than exports of other crops. Compared to other crops, tomatoes create a value that is around six times greater per unit of water.

**Table 4.** Economic water efficiency of agricultural exports of Egypt per crop during (2000–2021).

| | | | EWEX | | |
|---|---|---|---|---|---|
| **Year** | **Potato** | **Tomato** | **Onion** | **Rice** | **Cotton** |
| 2001 | 1.13 | 2.04 | 0.90 | 0.12 | 0.56 |
| 2002 | 1.24 | 2.26 | 0.94 | 0.12 | 0.52 |
| 2003 | 0.93 | 1.82 | 1.13 | 0.15 | 0.47 |
| 2004 | 1.18 | 2.19 | 1.52 | 0.19 | 0.73 |
| 2005 | 1.46 | 2.22 | 1.62 | 0.19 | 0.45 |
| 2006 | 1.34 | 2.48 | 1.55 | 0.21 | 0.62 |
| 2007 | 1.98 | 2.22 | 1.86 | 0.21 | 0.58 |
| 2008 | 3.54 | 3.30 | 4.82 | 0.31 | 0.42 |
| 2009 | 4.55 | 4.59 | 5.51 | 0.35 | 0.53 |
| 2010 | 3.16 | 3.69 | 3.71 | 0.25 | 0.55 |
| 2011 | 4.01 | 3.71 | 3.96 | 0.31 | 1.22 |
| 2012 | 3.62 | 6.95 | 4.30 | 0.38 | 0.47 |
| 2013 | 4.23 | 4.59 | 4.87 | 0.29 | 0.56 |
| 2014 | 3.82 | 5.02 | 4.21 | 0.26 | 0.75 |
| 2015 | 3.17 | 3.65 | 4.74 | 0.24 | 0.33 |
| 2016 | 3.32 | 5.15 | 3.55 | 0.40 | 0.60 |
| 2017 | 2.63 | 8.26 | 3.24 | 0.00 | 0.72 |
| 2018 | 2.20 | 15.76 | 2.68 | 0.00 | 0.69 |
| 2019 | 2.36 | 5.89 | 3.29 | 0.00 | 0.48 |
| 2020 | 2.98 | 5.30 | 3.24 | 0.00 | 0.39 |
| 2021 | 4.82 | 7.26 | 1.04 | 0.00 | 0.46 |

*3.4. Decomposition Analysis*

Decomposition analysis covers the period 2001–2021, which is consistent with Egypt-EU implementation of association agreement. Below in Table 5 are the results of the decomposition analysis for VW outflows for each crop and for the five crops collectively representing the agriculture sector. The period of 2001–2021 is divided into three parts: (1) 2001–2007 when virtual water exports declined from $3262 \times 10^5$ m³ to $2931 \times 10^5$ m³; (2) 2007–2014, when virtual water exports decreased sharply from $2931 \times 10^5$ m³ to $1510 \times 10^5$ m³; (3) 2014–2021, when virtual water exports dropped considerably from $1510 \times 10^5$ m³ to $454 \times 10^5$ m³. The rows below demonstrate the percentage contribution of scale, composition, and water intensity to this change.

The decrease on virtual water outflows in the first period is due to the reduction in cotton exports because of shrinking the cultivation area of cotton by 50 percent until it reached only 302.4 thousand hectares by the early 2000s, while the decrease in the second and third period is mainly due to the export ban imposed on rice exports, which represent the major consumer of irrigated water in the crops selected for this study. In total, VW outflows decreased by $2807.3 \times 10^5$ m³ between 2001 and 2021. In general, the composition effect has the most essential influence on the changes of VWX, accounting for at least 89%. The strong impact of this effect can be attributed to the significant changes in the structure of trade in Egypt over time. The contribution of water-intensive crops (i.e., rice and cotton) in the composition of trade has been declined noticeably, contributing to the reduction in virtual water outflows. Moreover, although potato, tomato, and onion composition of exports has increased, their contribution has a diminishing effect on virtual water outflows because of the reduction in their water footprint.

**Table 5.** Decomposition of Egyptian crops virtual water flows.

| Period | | 2001–2007 | 2007–2014 | 2014–2020 | 2001–2021 |
|---|---|---|---|---|---|
| Potato | DVW exports ($10^5$ m$^3$) | 93 | −153 | −8 | −68 |
| | scale effect (%) | 113 | 81 | 0 | −21 |
| | composition effect (%) | 24 | −10 | 95 | 204 |
| | water intensity effect (%) | −37 | 29 | 5 | −83 |
| Tomato | DVW exports ($10^5$ m$^3$) | 5 | −1 | −4 | 0 |
| | scale effect (%) | 8 | 284 | −16 | −170 |
| | composition effect (%) | 93 | −394 | 100 | 461 |
| | water intensity effect (%) | −2 | 210 | 16 | −191 |
| Rice | DVW exports ($10^5$ m$^3$) | 293 | −1179 | −1031 | −1917 |
| | scale effect (%) | 149 | 43 | 8 | 7 |
| | composition effect (%) | −27 | 55 | 96 | 89 |
| | water intensity effect (%) | −23 | 2 | −3 | 3 |
| Cotton | DVW exports ($10^5$ m$^3$) | −773 | −91 | −171 | −1035 |
| | scale effect (%) | −72 | 173 | 13 | −39 |
| | composition effect (%) | 170 | 26 | 25 | 140 |
| | water intensity effect (%) | 2 | −99 | 62 | −1 |
| Onion | DVW exports ($10^5$ m$^3$) | 52 | 2 | 159 | 213 |
| | scale effect (%) | 29 | −1926 | −140 | −13 |
| | composition effect (%) | 80 | 2910 | 204 | 181 |
| | water intensity effect (%) | −9 | −884 | 36 | −69 |
| All | DVW exports ($10^5$ m$^3$) | −330 | −1422 | −1055 | |
| | scale effect (%) | −288 | 55 | 4 | |
| | composition effect (%) | 357 | 46 | 94 | |
| | water intensity effect (%) | 30 | −1 | 2 | |

Note: Due to rounding errors, the effects do not always add up to 100%.

Despite a significant decrease in the average exported rice and cotton quantities, the water footprint of these crops increased by 3% and 15.5%, respectively. Unlike tomato, potato, and onion crops, their water footprints have decreased by an average of 16% for onions and tomatoes and 11% for potatoes over time, while export volumes have largely increased, especially for onions and tomatoes. This is reflected in the slight impact of the water intensity effect, which is positive for all crops between 2001–2007 and 2014–2021, also indicating its slight diminishing effect on VW of exports in the first and third period.

However, it's shown that potato, onion, and tomato export quantities have increased between 2001–2021. A reduction in the overall export quantities by 26% on average in the second period mitigated virtual water outflows and helped reduce the outflows of virtual water. This is highlighted by the magnitude of scale effect for all crops between 2007–2014.

## 4. Discussion

The study covers the period from 2001 to 2021 to focus on the most important crops exported to EU countries, which are also among the most important Egyptian crops. This period represents the start of the Association Agreement (AA) between Egypt and European Union (EU), which became effective in 2004. We divided the decomposition analysis into three periods.

- The first period (2001–2007) represents the date when Egyptian-EU agreement was signed.
- The second period (2008–2014) reflects the period of fully enacting the association and the complemented agreement on the liberalization of trade in agriculture and fisheries goods in 2010.
- The third period (2015–2021) demonstrates the period when the rice—top of Egyptian exports—trade ban was re-imposed in 2016 due to the increase of local prices and the reduction in rice self-sufficiency.

Egypt's export trends, export values, and VW outflows have changed over time. Since 2001, trade values are fluctuating and have a decreasing trend especially for cotton and rice causing the total export quantities to fall. A similar trend has been observed in fluctuating export quantities of vegetables for the same period. According to decomposition analysis, VW outflows have also decreased over time, primarily because of decreasing rice and cotton export volumes.

Although major crops were chosen to represent agricultural sector exports during Egypt-EU AA, the export curve trend is declining, particularly after 2008. This can be explained by the decline in cotton cultivation and the first rice export ban enacted in 2008, followed by some political tensions during the 25 January Revolution. On the other hand, this poor export performance was mirrored in a decrease in virtual water outflows, confirming the validity and significance of the rice trade ban as a strategic water-intensive commodity. The study emphasizes the importance of increasing potato, tomato, and onion exports due to their low water footprint, competitive prices, and high yield.

An earlier study by Torayeh explained that Egyptian vegetable and fruit exports face increased competition in EU markets. Additionally, the 2010 agreement affirms that Egypt is responsible for solving any problems, including sanitary and phytosanitary (SPS), and technical barriers to trade. The introduction of these measures, especially in recent years, coincided with a noticeable drop in Egyptian agricultural exports to the EU. Furthermore, Egypt's horticultural export competitiveness is hindered by the lack of adequate transportation. To import potatoes from Egypt, the EU required that every lot (or 25 tons) of Egyptian potatoes come from a "qualified area" and be subjected to harvesting, packaging, testing, and certification measures. As well, sanitary authorities in EU member states rejected other Egyptian agricultural supplies including oranges, coriander seeds, red pepper, onions, jams, and honeybees [51,52].

According to FAO, nearly 85% of all fresh exports are directed to Arab countries, over 12 percent of fresh exports go to Russia, while EU receives only 1.5%. This can be explained by several obstacles found mainly in Egypt's tomato trade market indicated below [53]:

- Methods of production in an open and unprotected cultivated land which causes low productivity and fails to meet international requirements, in addition to the absence of production technology.
- Lack of market orientation which arises from producer unawareness of different and highly profitable varieties demanded by the export market.
- Poor post-harvest handling due to quantitative and qualitative losses result from inadequate post-harvest infrastructure and logistics.
- Deficiencies in value chain governance resulting from the fragmentation of land.
- Insufficient safety and hygiene regulations because of inappropriate adopted agricultural practices.
- Unguaranteed profitability due to price fluctuation reduces incentives to invest in expensive value addition practices and technologies.

Moving to the sharp decline of rice and cotton exports, Agbo explained that Egypt's recent poor export performance on the international rice market is attributed to a rice export ban, poor coordination between the production and export sectors, a high percentage of losses and waste, a lack of quality standards, and an increase in Egyptian export prices when compared to export prices in competing states. As depicted in Figure 4, after 2001, a new rice subsidy program was implemented to increase Egyptian rice exports both in value and quantity. In response to rising domestic rice prices, an export ban on rice was imposed on 19 January 2008. As a result, Egypt was unable to sustain its exports and it declined sharply. This was accompanied with attempts to conserve scarce water resources by Egypt in 2009 through reducing land used for rice production. The Ministry of Industry and Foreign Trade removed the ban on Egyptian rice exports on 27 September 2012, under specific conditions and policies, increasing the volume and value of Egyptian rice exports in 2012 [54].

The ban was re-imposed in April 2016 due to price increases for most agricultural commodities, including rice. However, there was a decrease in rice self-sufficiency because of the adopted policy of reducing rice planted areas to address the problem of limited irrigation water [55].

In the late 19th and early 20th centuries, cotton accounted for Egypt's leading export, with an area dedicated to cotton cultivation peaking at 840 thousand hectares in 1961. In the 1960s and 1970s, Egypt produced about half a million tons of cotton a year. Cotton was subsidized by low administered prices on the local market, but at the same time, due to heavy taxation and rising wages, cotton cultivation was significantly reduced as farmers abandoned the crop for more profitable crops. Although area restrictions were lifted in the early 1990s, low profitability caused the total cotton area planted between 1980 and 2000 to shrink by 50 percent until it reached 302 thousand hectares by the early 2000s. Cotton was planted on only five percent of cultivated land in 2014, causing the government to stop subsidizing cotton in January 2015. Afterwards, the subsidy was reinstated at a different rate linked to the local spinning industry's selling price. Egypt's Ministry of Trade and Industry with the UN started a new project in July 2017 to improve the quality and performance of local cotton producers [45,56].

Accordingly, Egypt is saving water due to vegetable exports which have a much higher economic water efficiency than cereal and fiber exports. Due to the recent increase in exports of fruits and vegetables, Egypt is saving a large amount of water while selling at high prices, resulting in high economic water efficiency. Based on Egypt's trade experience, increasing exports of fruits and vegetables enhances the country's development strategy. However, it should consider the local water availability especially with the high dependance on irrigation from water resources originating outside the country's borders, growing population, and climate change impact. Therefore, it's crucial for decision makers to consider trade structure to sustain national water resources, this was confirmed by Zhong et al. study [57]. A policy supporting more efficient irrigation techniques or a switch to less water-intensive crops could achieve this. This has been confirmed by our decomposition analysis which emphasize the role of water efficiency on virtual water outflows.

A major limitation of the study is that it focused on the five main exported crops to the EU. A comprehensive study of virtual water trade is needed for the whole agriculture sector. This will provide a more comprehensive view of the link between international trade and water scarcity in Egypt, so that further studies in this area can be conducted. It was necessary to focus on these five crops because the meteorological data provided to estimate water footprint was for these specific territories, which entail high production for the crops selected in this study in these territories.

## 5. Conclusions

According to Hoekstra water assessment manual, Egypt is experiencing an urgent water crisis for a variety of reasons, including:

- The limited water resources.
- Approaching per capita water poverty line.
- Global warming, which may result in growing evapotranspiration for agricultural crops.
- The food security crisis for the growing population of strategic crops.
- Ethiopian Renaissance dam project.

The first goal of the study involved estimating virtual water flows of major crops contributing to agricultural trade with the EU. This has been accomplished through calculating WF of each crop and link it with long-term trade data. Unlike other studies of long-term trade trends, we used meteorological data from each territory station to calculate VW outflows accurately. To fulfil the second goal of the study, we analyzed the implications for virtual water outflows, economic water use efficiency, and conducted a decomposition analysis of the virtual water outflows.

Our findings provide insightful information for decision-makers. It suggests substituting exports of cotton and rice with onions, tomatoes, and potatoes. It also demonstrates

that research on virtual water trade should consider long-term dynamics. Egypt's agricultural trade market has high potential in the cultivation of fruits and vegetables, which recommends giving more attention to methods of production, post-harvest handling and logistics, and hygiene regulation.

Future research on the relationship between international trade and VW is encouraged to take a dynamic approach. It should look beyond national data by taking regional water scarcity into account. Furthermore, considering VW inflows to Egypt would also provide a comprehensive view of international trade's effect on water scarcity. This study describes an approach that may be helpful in addressing these issues. In nations suffering from water scarcity, the reduction of exports of intensive-water crops and the expansion of exporting crops with lower footprint is frequently viewed as an efficient way to promote economic growth, increase rural incomes, and reduce water poverty.

**Author Contributions:** Conceptualization, K.E.A.; Data curation, S.M.; Formal analysis, S.M.; Investigation, Y.S.; Methodology, S.M.; Project administration, K.E.A.; Resources, K.E.A.; Software, Y.S.; Supervision, K.E.A.; Validation, Y.S.; Writing—original draft, S.M.; Writing—review & editing, Y.S. All authors have read and agreed to the published version of the manuscript.

**Funding:** This research received no external funding.

**Institutional Review Board Statement:** Not applicable.

**Informed Consent Statement:** Not applicable.

**Data Availability Statement:** Data are available upon request to the corresponding author.

**Acknowledgments:** The authors would like to thank Hazim Mehawed, Samar El-Taher, and Aml Abo EL-Magd in agricultural engineering research center for their academic advice.

**Conflicts of Interest:** The authors declare no conflict of interest.

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
