# Peer review of "Decomposition Analysis of Virtual Water Outflows for Major Egyptian Exporting Crops to the European Union"

_sustainability, doi:10.3390/su15064943_

Round 1

Reviewer 1 Report

Recent data should be added. 

review the manuscript for spellcheck and grammatical mistakes.

Reviewer 2 Report

Dear authors,

Thanks for writing this manuscript. Studies regarding the crop’s virtual water and the water footprints of semi-desertic regions are of utmost importance to prevent future hazardous scenarios. However, manuscript requires modifications to enhance its understandability.

Firstly, the whole structure of the manuscript should be changed. It is not really clear what is the scientific advance. Why is it important to obtain the variation of the water footprint of five types of crops in a region of Egypt, if it has been dependent on government decisions? If conclusion points out that substitution of cotton and rice by tomato, potato or onions reduces the requirements of water, this is an obvious consequence. Suggesting different sort of crops depending on the measured evapotranspiration of the land, proposing an intelligent crop rotation alternative, or analyzing the evolution of the virtual water availability to prove the impossibility of harvesting certain crops in a region in the future due to climate change could be examples of possible targets. Historical progress of water footprints at a specific region, by itself, is more appropriate to a report and not to a scientific article. I recommend stating a proper study’s scientific objective very clearly in the abstract of the article, maintain it throughout the whole text, and ascertain it noticeably in the conclusion.

Continuing with the same idea, Introduction section seems to be quite extensive and it does not focus on any objective. It is more convenient that this section describes a problem, presents the difficulties to solve it, and at the end anticipates the possible solution proposed in the article.

As a matter of fact, Results section have too many references. This may denote that several parts of the article might have been overlapped. Comparison of the unpublished data shown in the article to previous published data is commonly presented in the Discussion section. Result’s one only exposes plainly the so far unrevealed measurements. By the way, Discussion section, as it is right now, resembles a Conclusion or part of it.

Another general concern entails evapotranspiration. Since it entails meteorological parameters, variation is expected. Perhaps, instead of using “average estimated values” (line 193), it would be useful knowing the evolution of this value through the twenty years' analyzed.

Besides, the following comments could be considered:

· It is recommended substituting the feddan unit to square meters or hectares.

· Notice that this article is expected to be understandable to people abroad Egypt. Maybe it could be convenient to describe at least once that “Upper and Lower Egypt regions” corresponds to Southern and Northern regions along the Nile river.

· Be aware of capital letters in middle words (i.e., in line 28: “Water”; this is not the only one).

· Be careful in describing acronyms before mentioning them (i.e., “WF” in line 60, “water footprint (WF)” in lines 63 and 64, and in line 68 it is again called without the acronym: ”water footprint”).

· There are too many references in this manuscript, but subsections 1.1, 1.2, 2.1, 2.3, and 2.5 seems to require even more. Consider that every data obtained from the literature might have its corresponding reference.

· Are Figure 5 and Table 4 both necessary? Perhaps only one is needed.

· In Table 5, accumulative corresponding to 2001-2020 seems to be missing.

· Check typing errors. For instance, in line 347, “Winter potatoes is […]”.

Best regards.

Reviewer 3 Report

A good study was done on "Decomposition analysis of virtual water outflows for major 2 Egyptian exporting crops to the European Union". Data complied Using CROPWAT 8.0 model, monthly reference evapotranspiration data from 2001 to 2020 were collected to calculate crop evapotranspiration and crop water use through Penman-Monteith equation for major crops exported to the European Union. All section was written well result and discussion portion written soundly. but some small observation from my side:

1. This article is written well but too lengthy, for reader interest, i can be do concise or shorten specially introduction part. 

2. Conclusion also too lengthy, it can also need to be shorten that will be more impressive 

3.  The authors should thoroughly check the reference throughout the text and cross check with reference for any disparity.

Overall, good article.

Reviewer 4 Report

The introduction section should be answering several questions: Why is the topic important (or why do you study it)? What are the research questions/objectives? What has been learned? What are your contributions? Why is to propose this particular method? I would suggest the author enhance the theoretical discussion and arrives at your debate or argument. The literature review section should be developed to motivate the readers in the background and currency of the subject of the research.  Clear research gaps should be discussed.

Round 2

Reviewer 2 Report

Dear authors,

Let me congratulate you for the work performed on the manuscript. It has improved considerably. I only have some minor comments, which might be considered as suggestions:

·  It is recommended to cite the original articles if they are mentioned, even if further studies or reviews are cited too. For instance, in lines 49 and 51 two articles are mentioned, but only the “review” reference number 19 is present. Equivalently in lines 149 and 446.

· Reference numbers should be revised: first one is the 7th, reference 27 is missing, 21 is mentioned in the section but there is no call for it, references from 22 to 26 are do not exist neither in the text nor in the last section, etc.

·  Be aware of the few typing errors left (for instance, dot instead of a comma in line 121 or first capital letter in line 126; or “EWE” in 379 line).

·  Figure 4 is repeated.

·  Description of the periods in the Discussion section might include the initial and final year of each period.

Best regards.
